

# Real-time loop-mediated isothermal amplification for rapid detection of *Enterocytozoon hepatopenaei*

Shao-Xin Cai[1,2], Fan-De Kong[2], Shu-Fei Xu[2] and Cui-Luan Yao[1]

[1] Fisheries College of Jimei University, Xiamen, China
[2] Xiamen Customs District/State Key Laboratory for Crustaceans Quarantine, Xiamen, China

## ABSTRACT

**Background:** *Enterocytozoon hepatopenaei* (EHP) is a newly emerged microsporidian parasite that causes retarded shrimp growth in many countries. But there are no effective approaches to control this disease to date. The EHP could be an immune risk factor for increased dissemination of other diseases. Further, EHP infection involves the absence of obvious clinical signs and it is difficult to identify the pathogen through visual examination, increasing the risk of disease dissemination. It is urgent and necessary to develop a specific, rapid and sensitive EHP-infected shrimp diagnostic method to detect this parasite. In the present study, we developed and evaluated a rapid real-time loop-mediated isothermal amplification (real-time LAMP) for detection of EHP.

**Methods:** A rapid and efficient real-time LAMP method for the detection of EHP has been developed. Newly emerged EHP pathogens in China were collected and used as the sample, and three sets of specificity and sensitivity primers were designed. Three other aquatic pathogens were used as templates to test the specificity of the real-time LAMP assay. Also, we compared the real-time LAMP with the conventional LAMP by the serial dilutions of EHP DNA and their amplification curves. Application of real-time LAMP was carried out with clinical samples.

**Results:** Positive products were amplified only from EHP, but not from other tested species, EHP was detected from the clinical samples, suggesting a high specificity of this method. The final results of this assay were available within less than 45 min, and the initial amplification curve was observed at about 6 min. We found that the amplification with an exponential of sixfold dilutions of EHP DNA demonstrated a specific positive signal by the real-time LAMP, but not for the LAMP amplicons from the visual inspection. The real-time LAMP amplification curves demonstrated a higher slope than the conventional LAMP.

**Discussion:** In this study, pathogen virulence impacts have been increased in aquaculture and continuous observation was predominantly focused on EHP. The present study confirmed that the real-time LAMP assay is a promising and convenient method for the rapid identification of EHP in less time and cost. Its application greatly aids in the detection, surveillance, and prevention of EHP.

Corresponding authors
Shu-Fei Xu, xusf@xmciq.gov.cn
Cui-Luan Yao, clyao@jmu.edu.cn

## INTRODUCTION

*Enterocytozoon hepatopenaei* (EHP) is the microsporidian parasite that causes *hepatopancreatic microsporidiosis* in shrimp (*Chayaburakul et al., 2004*; *Tangprasittipap et al., 2013*). Moreover, EHP is an emerging pathogen that affects the cultured shrimp *Penaeus vannamei* in many countries, such as Vietnam, Thailand, Brunei, Malaysia, Indonesia, China, and Venezuela (*Chayaburakul et al., 2004*; *Liu et al., 2017*; *Rajendran et al., 2016*; *Tang et al., 2015*, *2016*; *Tangprasittipap et al., 2013*; *Thi Ha et al., 2010*), and is imposing a continuous threat to shrimp farming industries in future. The shrimp with EHP could be an immune risk factor, causing an increased susceptibility to pathogens of both acute hepatopancreatic necrosis disease and septic hepatopancreatic necrosis (*Aranguren, Han & Tang, 2017*).

*Enterocytozoon hepatopenaei* is an intracellular parasite that replicates within the cytoplasmic area of the tubule epithelial cells in the hepatopancreas, which was considered to be a new species, *E. hepatopenaei*, within the genus *Enterocytozoon* (*Tourtip et al., 2009*). Further, EHP infection involves the absence of obvious clinical signs and it is difficult to identify the pathogen through visual examination, increasing the risk of disease dissemination. To control EHP, the EHP-infected broodstock from the cultivation system is excluded. However, it is difficult to perceive EHP due to the absence of obvious clinical symptoms when the shrimp was infected. Hence, it is urgent and necessary to develop a specific, rapid, and sensitive EHP-infected shrimp diagnostic method to detect this parasite.

It is well known that modern molecular biology detection techniques play an important role in pathogen detection, and there have been major improved techniques, such as PCR assay, in situ hybridization, and real-time PCR put forward in recent years (*Cai et al., 2012*). But due to expensive equipment, and reagents limited their broader application. Also, histological observations were limited due to the individual size of EHP, and it remained difficult to detect some small individual (*Tourtip et al., 2009*). While in situ hybridization has a low sensitivity and may lead to few false-positive results. Consequently, a rapid detection method for EHP is necessary for suitable handling, prevention of spreading, and reducing the risk factors caused by this pathogen.

In order to develop a faster, more reliable and more cost-effective assays, we applied a loop-mediated nucleic acid isothermal amplification (LAMP) to develop a real-time loop-mediated isothermal amplification (real-time LAMP) for detecting EHP. The real-time LAMP assay is a more sensitivity and accurate diagnostic method, which enables the detection of low-level infections and infection in the host. This assay remained a critical tool to detect EHP in shrimps.

## MATERIALS AND METHODS

### Animal samples

Four EHP positive shrimp samples are naturally infected were collected from the shrimp ponds in Zhanjiang, Guangdong, China in February 2017. All the positive samples were confirmed by the state key laboratory for crustacean quarantine. Clinical samples

**Table 1 Reference primers used in the study.**

| Name | Sequences (5′→3′) | Length (nt) |
|---|---|---|
| **Real-time LAMP** | | |
| EHP-OF | AGGTGGGCAAAGAATGAAAT | 20 |
| EHP-OB | AAGCAGCACAATCCACTC | 18 |
| EHP-IF | CCCAGCATTGTCGGCATAGTATCAAGGACGAAGGCTAGA | 39 |
| EHP-IB | TGTTGCGAGAGCGATGCTCCTTGCGAGCGTACTATC | 36 |
| EHP-LF | AGAACTACAGCGGTGTCTAATC | 22 |
| EHP-LB | TGGTGTGGGAGAAATCTTAGTT | 22 |
| **LAMP** | | |
| EHP-F3 | GGGATCAAGGACGAAGGCT | 19 |
| EHP-B3 | GGGATCAAGGACGAAGGCT | 19 |
| EHP-FIP | AAGCATCGCTCTCGCAACACCACACCGCTGTAGTTCTAGCA | 41 |
| EHP-BIP | TTCGGGCTCTGGGGATAGTACGGTCCTTCCGTCAATTTCGCT | 42 |

were collected from the shrimp ponds in Zhanjiang, Guangdong, China. Then, the dissected hepatopancreatic tissues of the shrimp were taken and preserved separately using 95% ethanol with more than three times the volume. The White spot syndrome virus (WSSV), Macrobrachium rosenbergii Nodavirus (MrNV), and Infectious hypodermal and hematopoietic necrosis virus (IHHN) used in the present study was provided by the research teams at state key laboratory for crustaceans quarantine.

## DNA extraction and template preparation

The hepatopancreatic tissues were rinsed under sterile water to remove ethanol. This was followed by mixing of 50 mg–1 g of shrimp hepatopancreatic tissues with two times the volume of lysis buffer (100 mM pH 8.0 Tris–HCl, one mM pH 8.0 EDTA, 1% Sodium Dodecyl Sulfate (SDS) and sterile water) at room temperature for 5 min, and then lysed at 95 °C for 5 min. The mixture was transferred to the DNeasy Mini spin column and centrifuged for 1 min. Serial dilutions of EHP DNA ($10^0$ to $10^{-6}$) were prepared using Tris buffer (TE, pH 8.0), and then stored at 20 °C until further use.

## Oligonucleotides for real-time LAMP

The real-time LAMP primers targeting EHP were selected from the established methods for this study (Table 1). The real-time LAMP primers (EHP-OF, EHP-OB, EHP-IF, EHP-IB, EHP-LF, EHP-LB) and LAMP primers were designed using Primer Explorer ver. 4 (https://primerexplorer.jp/elamp4.0.0/index.html). These primers were synthesized by Sangon Biotech Co., Ltd (Shanghai, China).

## Real-time LAMP reaction

The real-time LAMP reaction was performed in a total volume of 25 μl, including 0.5 μl (8U) of Bst DNA Polymerase Large Fragment (NEB, Beijing, China), four μM of SYTO-9 (Thermo Fisher Scientific, Shanghai, China), one μl EHP-OF & EHP-OB (0.2 μmol/l), one μl EHP-IF & EHP-IB (1.6 μmol/l), one μl EHP-LF & EHP-LB (0.8 μmol/l), one μl of

EHP template DNA, and 13 µl of real-time LAMP buffer. The real-time LAMP buffer contains 20 mM Tris–HCl, 10 mM KCl, 10 mM $(NH_4)_2SO_4$, two mM $MgSO_4$, 0.1% of Triton X-100, 1.4 mM dNTPs (TaKaRa, Dalian, China) each, and sterile water was used to make up to 25 µl. The fluorescent signals were observed automatically by StepOne Real-Time PCR System (ABI, Shanghai, China).

## LAMP reaction

The LAMP reaction was performed in a total of 25 µl volume, including 0.5 µl (8U) of Bst DNA Polymerase Large Fragment (NEB, Beijing, China), four µM of SYTO-9 (Thermo Fisher Scientific, Beijing, China), one µl, 12.5 µl of LAMP buffer (containing 20 mM Tris–HCl, 10 mM KCl, 10 mM $(NH_4)_2SO_4$, two mM $MgSO_4$, 0.1% of Triton X-100, primer mix (EHP-F3 0.2 µM, EHP-B3 0.2 µM, EHP-FIP 1.6 µM, EHP-BIP 1.6 µM), two µl EHP template DNA, 1.4 mM dNTPs each), and sterile water was used to make up to 25 µl. The reaction was carried out at 60 °C for 45–90 min.

## Specificity of real-time LAMP detection

To determine the specificity, the established real-time LAMP method was carried out using different sources of DNA isolated from WSSV, MrNV, IHHN, and with EHP template DNA, $ddH_2O$ was used as a negative control. Also, clinical samples were diagnosed with the established real-time LAMP method, the EHP DNA was used as a positive sample, and $ddH_2O$ was used as a negative control.

## The sensitivity of real-time LAMP

The sensitivity of real-time LAMP procedure developed was compared with LAMP, and the primers used in the real-time LAMP and LAMP were listed in Table 1. Serial dilutions of EHP DNA ($10^0$ to $10^{-6}$) were used to be a positive sample, and $ddH_2O$ was used as a negative control. The real-time LAMP the fluorescence signal date was obtained from StepOne Real-Time PCR System.

## Comparison with the conventional LAMP

The EHP DNA was used as the template, and $ddH_2O$ was used as a negative control. The primers used in the real-time LAMP and LAMP were listed in Table 1. The reactions were carried out under the conditions mentioned above. The real-time LAMP was compared with the conventional LAMP by their amplification curves. The fluorescent signals were observed automatically by StepOne Real-Time PCR System.

# RESULTS

## The established real-time LAMP primers are tested

The real-time LAMP was performed with four different EHP samples and $ddH_2O$ as a control template to test the set of primers. The S type curve demonstrated a positive result, while the smooth straight line a negative result. From the appearance of the amplification curves, the set of primers demonstrated increased fluorescent signals without any amplification peak in the negative control (Fig. 1).

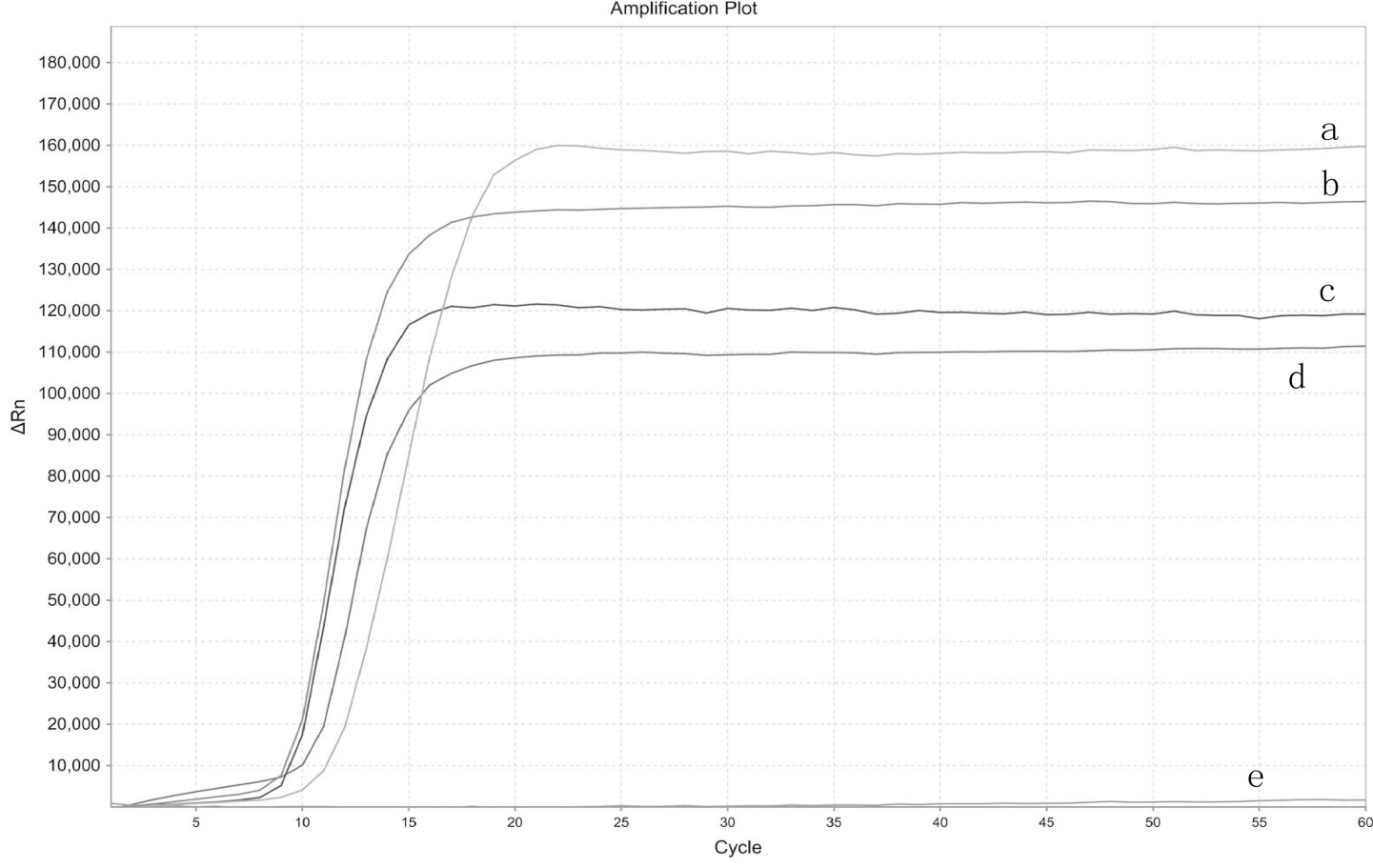

**Figure 1 Primers tested for real-time LAMP.** The amplification curve graph was plotted automatically by StepOne Real-Time PCR System. ΔRn is the fluorescence unit minus the baseline, the graph depicts the ΔRn on the Y-axis and cycle number on the X-axis. (A) EHP1. (B) EHP 2. (C) EHP 3. (D) EHP 4. (E) Negative control.

## The established real-time LAMP shows high specificity

To test the specificity of real-time LAMP assay, three other aquatic pathogens were used as templates. The cycle number at which the amplified products surpassed the threshold of detection was observed as the positive fluorescent signal. Along with EHP DNA, various pathogens including WSSV, MrNV, and IHHN have been tested. Positive products were amplified only from EHP, and the curves of the other three tested species showed straight lines, suggesting no amplification (Fig. 2). Also, EHP was detected in clinical samples (data no-show). The specificity test of real-time LAMP revealed that the assay developed showed no cross-reactivity with other aquatic pathogens, indicating that the method was highly specific.

## The established real-time LAMP is highly sensitive

Sensitivity analysis was performed using a 10-fold serial dilution set of template DNA of the above mentioned EHP. Serial 10-fold dilutions ranging from $10^0$ times to $10^{-6}$ times of EHP positive sample DNA were tested by real-time LAMP and LAMP. We found that the amplification with an exponential of sixfold dilutions of EHP DNA demonstrated

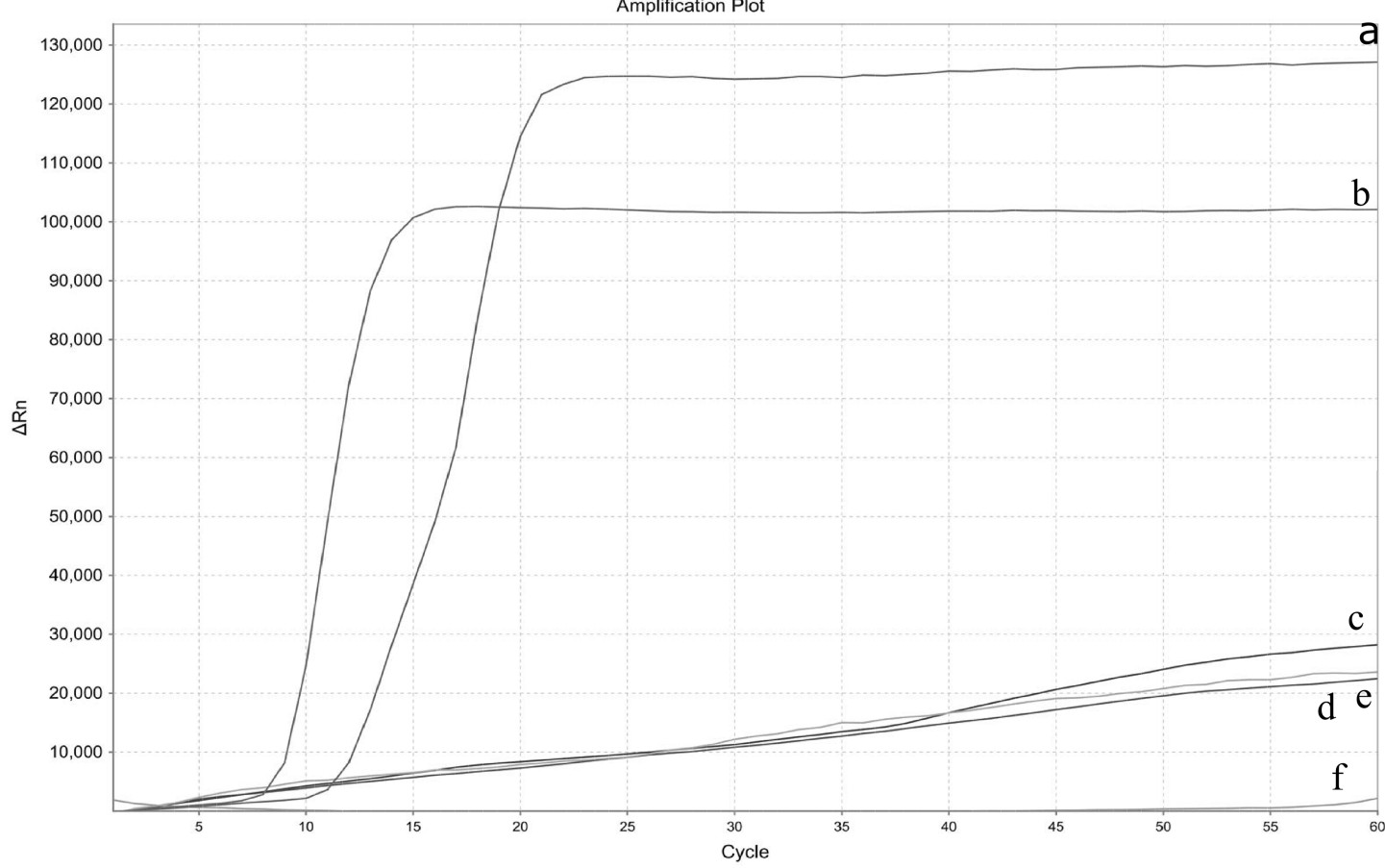

**Figure 2 Specificity test of the real-time LAMP for *Enterocytozoon hepatopenaei*.** Three different pathogen templates were performed along with *Enterocytozoon hepatopenaei* (EHP). (A) EHP 3. (B) EHP 4. (C) White spot syndrome virus (WSSV). (D) *Macrobrachium rosenbergii* noda virus (MrNV). (E) Infectious hypodermol and hematopoietic necrosis virus (IHHN). (F) Negative control.

a specific amplification curve by the real-time LAMP (Fig. 3). From the visual inspection of the LAMP amplicons, the turbidity change of the tube was clearly observed during amplification with an exponential of fivefold dilutions of EHP DNA, but not for the exponential of sixfold dilutions of EHP DNA (Fig. 4). This indicated that the sensitivity of the real-time LAMP assay was higher than the LAMP.

## The established real-time LAMP has high efficiency

The real-time LAMP has added a pair of loop-primer compared to the conventional LAMP. The efficiency was compared with the LAMP using their amplification curves. The cycle number at which the amplified products surpassed the threshold of detection could be observed as the positive fluorescent signal. As shown in the figure, real-time LAMP amplification curves demonstrated a higher slope than the conventional LAMP. It took more than 60 cycles for the LAMP to reach the amplification plot, while 14 cycles of amplification for the real-time LAMP to reach the same level (Fig. 5). These results revealed that the real-time LAMP demonstrated more amplification efficiency than the conventional LAMP.

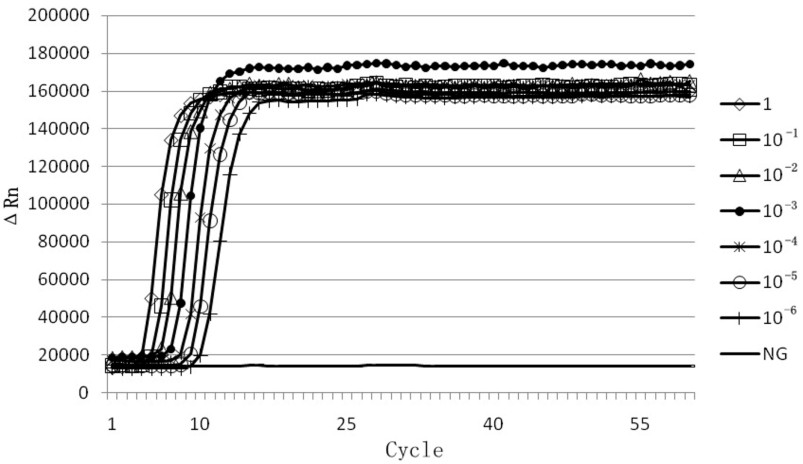

**Figure 3 Sensitivity of real-time LAMP for *Enterocytozoon hepatopenaei*.** The fluorescence unit vs. cycle graph was plotted from the data of the StepOne Real-Time PCR System. Serial 10-fold dilutions ranging from $10^0$ times to $10^{-6}$ times for *Enterocytozoon hepatopenaei* (EHP) positive sample DNA. (NG) Negative control.

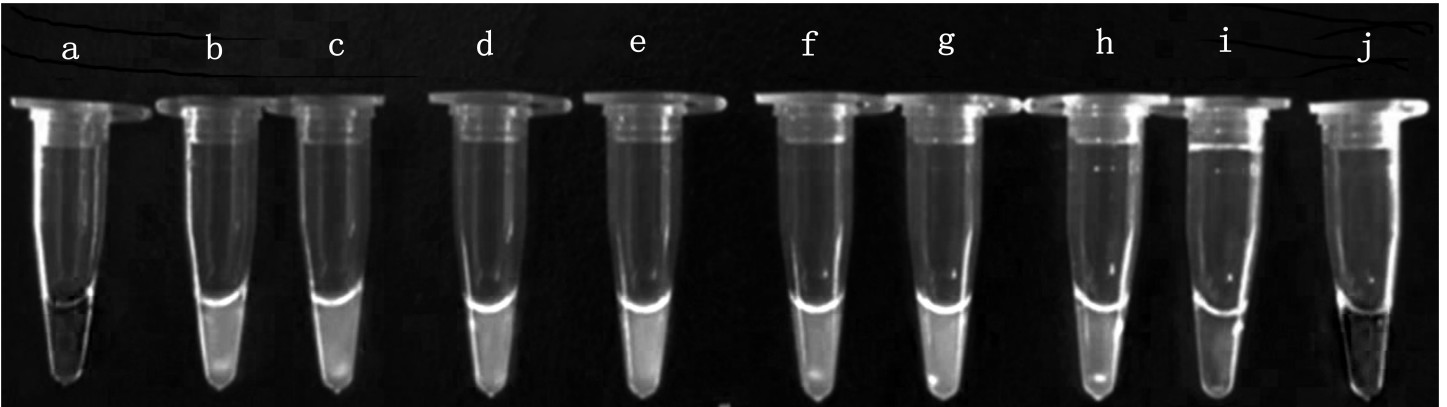

**Figure 4 Visual inspection of the LAMP amplicons for *Enterocytozoon hepatopenaei*.** (A) Negative control. (B) Positive control. (C) *Enterocytozoon hepatopenaei* (EHP). (D–I) Corresponding to serial 10-fold dilutions ranging from $10^0$ times to $10^{-6}$ times for EHP positive sample DNA, respectively. (J) Negative control.

## DISCUSSION

*Enterocytozoon hepatopenaei* is a newly emerged microsporidian parasite that causes retarded shrimp growth in cultures of many shrimp farming countries, but no effective approaches to control this disease has been developed to date. The pathogens of EHP are hard to eliminate. Complete inhibition of the activity of EHP spores is demonstrated either by freezing the spores at −20 °C for at least 2 h or by treating them with chemicals (*Aldama-Cano et al., 2018*). It is difficult to distinguish the EHP infected ones with normal ones, and many molecular detection methods have been developed to detect the pathogens.

Recently, many works have been done to detect EHP-infected shrimp, which included histological observation, PCR assay and in situ hybridization (*Tangprasittipap et al., 2013*),

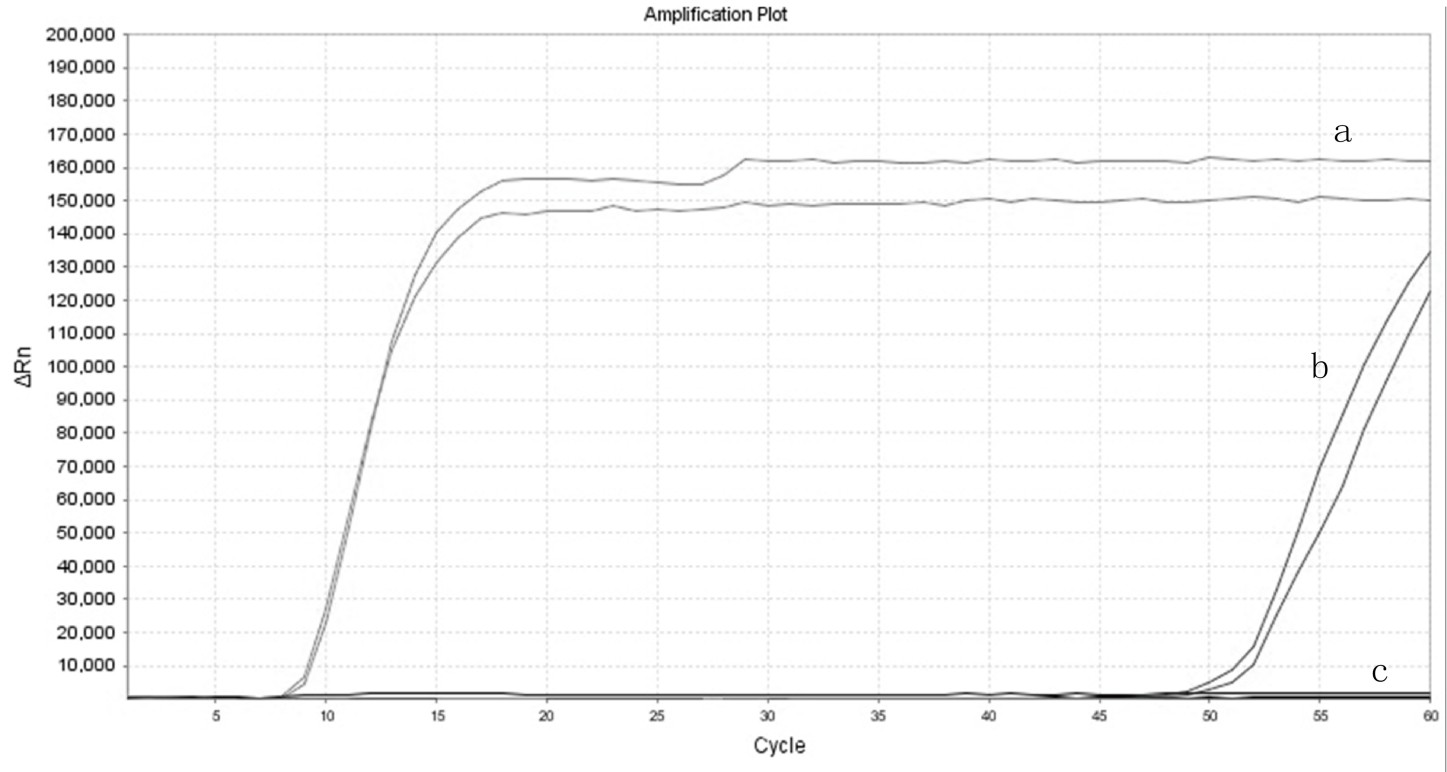

**Figure 5 The amplification efficiency of real-time LAMP compared to the LAMP.** (A) Real-time loop-mediated isothermal amplification. (B) Conventional loop-mediated isothermal amplification. (C) Negative control.

real-time PCR (*Liu et al., 2017*). These techniques require complicated equipment such as thermocycler, electrophoresis set, gel documentation system and trained personnel, and could not be used for on-site detection.

The LAMP technique is a rapid, sensitive and specific detection method for the pathogen (*Jelocnik et al., 2017*), and is also applied to detect the parasites, such as toxoplasmosis (*Lau et al., 2010*). Therefore, the efficiency of the LAMP is more sensitive than normal PCR. The amplification products of LAMP assays are visually detected by observation (*Goo et al., 2016*), but the results were always affected by distinctiveness, the results are obtained only after the reaction has been finished, and it cannot distinguish high concentration or low concentration DNA sample.

The real-time LAMP is a new DNA amplification technique. The real-time LAMP was more sensitive than the nested PCR (*Sritunyalucksana et al., 2006*). Compared to the LAMP, the real-time LAMP has added a set of loop primers, which increased the efficiency of amplification and made the detection more rapidly. The positive results were seen as a curve through the screen directly (*Karthikeyan et al., 2017*). Low-level infections of DNA pathogens can also be detected (*Mori et al., 2004*; *Notomi et al., 2000*).

The target sequence can be amplified under isothermal conditions with high efficiency, rapidity, and specificity. Positive products were amplified only from EHP, but not from any other tested species. Serial dilution test showed that the real-time LAMP was more sensitive than the LAMP. Because of higher amplification efficiency, the real-time

LAMP can save almost half the time than the LAMP. The use of specific loop primers for detecting EHP can reduce the necessary amplification time to less than 45 min, the initial amplification curve was observed from about 6 min.

In addition, the assay is easy to operate and determine, performed under isothermal conditions, and can be used by people without any operation experience. The tube lid need not be opened to eliminate the aerosol pollution for excluding the false positive results. While the LAMP product was detected by white precipitate or color change, which is not easy to detect (*Deb et al., 2016*). It can greatly improve the detection sensitivity and specificity.

## CONCLUSIONS

In this study, pathogen virulence impacts have been increased in aquaculture and continuous observation was predominantly focused on EHP. The present study confirmed that the real-time LAMP assay is a promising and convenient method for the rapid identification of EHP in less time and cost. Its application greatly aids in the detection, surveillance, and prevention of EHP.

## ACKNOWLEDGEMENT

We thank associate Prof. Ming-sheng CAI for the helpful advice on assays.

### Funding
This work was supported by the Science and Technology Program of Jimei (20172C01). The funders had no role in study design, data collection and analysis, decision to publish, or preparation of the manuscript.

### Grant Disclosure
The following grant information was disclosed by the authors:
Science and Technology Program of Jimei: 20172C01.

### Competing Interests
The authors declare that they have no competing interests.

### Author Contributions
- Shao-Xin Cai conceived and designed the experiments, performed the experiments, analyzed the data, contributed reagents/materials/analysis tools, prepared figures and/or tables, authored or reviewed drafts of the paper, approved the final draft.
- Fan-De Kong practical application of the real-time LAMP to clinical samples.
- Shu-Fei Xu conceived and designed the experiments, performed the experiments, analyzed the data, contributed reagents/materials/analysis tools, prepared figures and/or tables, authored or reviewed drafts of the paper.
- Cui-Luan Yao approved the final draft.

## DNA Deposition

The following information was supplied regarding the deposition of DNA sequences:

GenBank, accession number: MNPJ01000014.1.

## Data Availability

The raw measurements are provided in the Supplementary Files.

## Supplemental Information

Supplemental information for this article can be found online at http://dx.doi.org/10.7717/peerj.5993#supplemental-information.

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
