# Peer review of "Real-time loop-mediated isothermal amplification for rapid detection of Enterocytozoon hepatopenaei"

_PeerJ, doi:10.7717/peerj.5993_

## Round 0.1 · original submission · Minor Revisions

Thank you for submitting your manuscript to PeerJ. It has now been reviewed by experts in the field who commented favorably on the significance of the work to the field. All had comments to enhance the manuscript as below. It should also be polished by a native speaker.

Reviewer 1 ·

Basic reporting

'no comment'

Experimental design

The authors in this paper were aimed to establish an effective diagnostic method for the detection and prevention of Enterocytozoon hepatopenaei (EHP), which has emerged as a serious pathogen reported to be associated with retarded growth in cultured shrimp. The real time loop-mediated isothermal amplification (real-time LAMP) for rapid detection of EHP has been developed. This method was compared with the conventional LAMP, showing less time and cost consumption. Its application greatly aids in the surveillance, prevention, and control of EHP. Thus the method for EHP presented might mean a contribution to the cultured shrimp industry. For benefit of the reader, however, before acceptance for publication, I would recommend minor changes to be taken into account in the manuscript.

Validity of the findings

'no comment'

Additional comments

Recommendation: The paper is probably publication, but the paper should be reviewed again in revised form before it is accepted.
Additional Comments: Firstly, regarding the EHP, I think the following article should be cited in the chapter introduction:
Tourtip, S., Wongtripop, S., Stentiford, G.D., Bateman, K.S., Sriurairatana, S., Chavadej, J.,Sritunyalucksana, K., Withyachumnarnkul, B., 2009. Enterocytozoon hepatopenaei sp. nov. (microsporida: Enterocytozoonidae), a parasite of the black tiger shrimp Penaeus monodon (Decapoda: Penaeidae): fine structure and phylogenetic relationships. J. Invertebr. Pathol. 102, 21–29.
Secondly, the labeling in Figure 4 should be lowercase.
Thirdly, I think that there are words more appropriate other than “primers are preferred” in the result in line 165.
Fourthly, the paper also has some typos and language issue which needs to be checked and corrected in the revision.

Reviewer 2 ·

Basic reporting

no comment

Experimental design

Please provide the data about practical application to clinical samples.

Validity of the findings

no comment

Additional comments

Enterocytozoon hepatopenaei (EHP) is a newly emerged microsporidian parasite that causes retarded shrimp growth. In this study, the authors established and evaluated a rapid real-time Loop-Mediated Isothermal Amplification (real-time LAMP) for detection of EHP. It is important and necessary to develop a specific, rapid and sensitive EHP diagnostic method to identify this parasite. However, I think the authors should consider the following comments:

(1) EHP infection involves the absence of obvious clinical signs and it is difficult to identify the pathogen through visual examination. Then in the field, in order to control or eliminate EHP, how were samples collected from the shrimps without clinical signs? Does sampling cause enormous damage to shrimp?
(2) In this study, a real-time LAMP for detection of EHP were developed. But in the parts (Materials and methods, and Results), there is not data about the practical application of the real-time LAMP to clinical samples. Please provide the relative data.
(3) On lines 209-210, are you sure that the equipment for the LAMP is not expensive?
(4) The real-time LAMP is helpful to identify the EHP infection. But it is difficult to control parasite in the shrimps using this method. Some drugs or vaccines might eliminate EHP. Please revised the manuscript.

Reviewer 3 ·

Basic reporting

The Enterocytozoonhepatopenaei(EHP) becomes more prevalent pathogen since the pathogen was associatedwith retarded growth. Therefore an effective detection method for the pathogen is necessary. This MS have developed real-time loop-mediated isothermal amplification (real-time LAMP) for EHP, which is very helpful for survey and detection.

Experimental design

no comments

Validity of the findings

the findings of the MS is already validated.

Additional comments

1. The introduction at the first of the paper is too simple for the readers, as a newly microsporidian parasite, it is the characteristics of this article. I believe that it will be helpful if the authors can make a more detail.
2. EHP is a pathogen, not a disease, for the word “EHP diagnostic method”in line 36, so maybe should add a word “infected” to make it more clearly, “EHP-infectedshrimp diagnostic method”.
3. There are four pathogens used in the paper, so the word “pathogen” in line 40 should be “pathogens”, also the “was” should be “were”correspond.
4. The word “EHP diagnostic” in line 91 should be “EHP-infected shrimp”.
5. The first sentence in the part “The established real-time LAMP is highly sensitive’ in line 182 should be adjusted to “Sensitivity analysis was performed…”.
6.For the legend of Figure 2, there are many repeated descriptions“Amplification curve” in line269-272, the legend about different curve could short for “(A)EHP3.(B) EHP

---

## Round 0.2 · accepted · Accept

Your revision has addressed the concerns. Please try your best to generalize this rapid real-time Loop-Mediated Isothermal Amplification (real-time LAMP) method in detection of EHP.

#